# Analytical Validation of the IMMULITE^®^ 2000 XPi Progesterone Assay for Quantitative Analysis in Ovine Serum

**DOI:** 10.3390/ani12243534

**Published:** 2022-12-14

**Authors:** Kristi L. Jones, Ameer A. Megahed, Brittany N. Diehl, Ann M. Chan, Oscar Hernández, Catalina Cabrera, João H. J. Bittar

**Affiliations:** 1Department of Large Animal Clinical Sciences, College of Veterinary Medicine, University of Florida, Gainesville, FL 32610, USA; 2Department of Animal Medicine (Internal Medicine), Faculty of Veterinary Medicine, Benha University, Moshtohor-Toukh, Kalyobiya 13736, Egypt; 3Department of Research and Graduate Studies, College of Veterinary Medicine, University of Florida, Gainesville, FL 32610, USA; 4Department of Small Animal Clinical Sciences, College of Veterinary Medicine, University of Florida, Gainesville, FL 32610, USA

**Keywords:** sheep (ovine), progesterone, analytical performance validation

## Abstract

**Simple Summary:**

Consumer demand for sheep food products are on the rise, thus resulting in shifts in sheep production systems. These shifts require a better understanding of reproductive health in various production systems through research and implementation of these improved reproductive management strategies. To do this it is necessary to utilize safe, timely and accurate tools to measure key reproductive markers. Circulating progesterone concentration is a commonly used reproductive marker in ewes as it can be used to monitor female cyclicity. We evaluated how well the IMMULITE^®^ 2000 XPi (Siemens) measured progesterone concentrations in sheep serum. This system is optimized for use in human clinical settings, and we know that the sample type can influence system performance therefore our objective was to determine the analytical performance of the machine using sheep serum samples spiked with known concentrations of progesterone over the manufacturers reported readable range. Experiments were conducted in accordance with guidelines recommended by the Quality Assurance and Laboratory Standard committee of the ASVCP. Variability within and between runs as well as on independent machines fell within acceptable deviation, therefore we concluded that use of the IMMULITE^®^ 2000 XPi (Siemens) is a safe, timely and accurate system to measure circulating progesterone concentrations in sheep serum.

**Abstract:**

Monitoring circulating progesterone (P4) concentration is an important component of basic and applied reproduction research and clinical settings. IMMULITE^®^ 2000 XPi (Siemens) is a newly upgraded fully automated immunoassay system marketed for human use to measure concentrations of different analytes including P4. Our objective was therefore to characterize the analytical performance of the IMMULITE^®^ 2000 XPi P4 immunoassay across the reportable range in ovine serum. This validation of analytical performance included determining (1) linearity, (2) precision through within-run, and between-run coefficient of variation (CV) calculations, (3) accuracy through bias calculations for spiking-recovery bias and interlaboratory (range and average based) bias for two laboratories across the reportable range (0.2–40 ng/mL). The average within-run and between-run precision (CV%) across the reportable range of the IMMULITE^®^ 2000 XPi P4 immunoassay for serum P4 concentration were both <5%, ranging between 2–8%. The average Observed Total Analytic Error (TEo) reported here for serum P4 concentration across the reportable range was ~30%, ranging from 14.8–59.4%, regardless of the considered bias. Based on these data we conclude that the automated IMMULITE^®^ 2000 XPi P4 immunoassay provides a precise, accurate, reliable, and safe method for measuring P4 concentration ovine serum.

## 1. Introduction

Progesterone (P4) is a conserved steroid hormone [1] and concentration of P4 plays a role in both establishment and maintenance of pregnancy [2,3,4]. Circulating P4 can be measured in serum or plasma and levels are both cyclical and age dependent. Specifically, in female sheep, >1 ng/mL circulating P4 concentration indicates luteal phase of cycling [5]. Therefore, circulating P4 concentration can be used to determine puberty and inform reproductive management practices in female small ruminants [6]. As sheep extensive production systems increase to aid in global food security, multiple challenges including reproduction need to be studied and addressed to best optimize sustainability of these systems [7]. Timed artificial insemination and embryo transfer are some examples of methods used in reproduction management, and it has been shown that circulating P4 concentration at various times can influence the success of these techniques in sheep [8]. As research and reproductive management in small ruminants progresses it is essential that methods used to measure P4 concentration in blood components are reliable, accurate, safe, timely, and cost effective. The historical lab standard to measure circulating P4 levels is radio-immunoassay, however, its use is limited by time constraints, cost, safety and regulatory concerns [9]. Therefore, the use of colorimetric, chemiluminescent based immunoassay (CLEIA) has been increasing, and fully automated CLEIA systems are now widely used in human clinical settings to measure circulating P4 concentration. The IMMULITE^®^ 2000 XPi (Siemens, Cary, NC, USA) is one of these automated systems. It is a closed system designed to measure a variety of analytes, including P4 in a multiplex array from multiple bodily fluid sources. This machine has been designed and validated for use in human samples. However, although there are reports of using the IMMULITE^®^ 2000 XPi, and its predecessor the IMMULITE^®^ 1000 to measure circulating P4 concentration in veterinary setting [10,11,12], there are no comprehensive analytical performance reports addressing the accuracy or reliability of this data in ruminants. This is especially important as reproductive and clinical demands are increasing, specifically in small ruminants as a result of progressive producer interest in breeding practices. 

It is essential to know caveats and analytical performance limitations of any assay used to detect analytes in both research and clinical settings. Researchers, clinicians and laboratory technicians need to work together to understand physiological relevance and limitations of detection assays. Analytical quality of an assay can be evaluated based on the concept of observed total analytic error (TEo) [13] which takes the system as a whole into consideration. Westgard and colleagues (1974) [14] described the concept of TEo as a combination of imprecision (random error: variability between different measurements of the same analyte) and bias (inaccuracy or systematic error: skewness of measurements from the true value). Sources of bias include the machine, technique, technician and sample. The fact that sample contributes to the observed total error makes it necessary to test the analytical parameters of this method on sheep serum despite the company providing analytical performance data for the machine using human samples. The American Society for Veterinary Clinical Pathology guidelines [15] accepts an analyte detection method for use in clinical settings when TEo < allowable total error (TEa); where TEa is the maximum analytical error tolerable and still detect clinically useful differences in results. TEa is informed by research and determined through a clinical consensus discussion. Like TEo, TEa is dependent on several factors including, species, analyte concentration, clinical use, or type of laboratory [16]. To date the recommended TEa is not available for veterinary endocrinology [17]. In the absence of TEa information, analytical performance parameters can provide critical information on the reliability of data obtained from an analyte detection method [18].

Due to the conservation of P4 across mammalian species and use in human clinical settings we hypothesized the IMMULITE^®^ 2000 XPi will provide an accurate and reliable method for measuring serum and plasma P4 concentration in sheep samples. Therefore, the objective of our study was to experimentally determine analytical performance parameters of the IMMULITE^®^ 2000 XPi as it applies to measuring serum P4 concentration from sheep samples. This information is intended to independently inform research and clinical interpretation of sheep serum P4 concentration data obtained using the IMMULITE^®^ 2000 XPi closed system following default manufacturer protocol.

## 2. Materials and Methods

### 2.1. Ethics

The serum samples used for these experiments were collected in accordance with ethical practices. Pooled serum samples for progesterone free spiking samples were collected as part of a teaching lab and practices were approved by University of Florida IACUC. Patient samples used for the inter-lab analysis were collected as part of a collaborative project to examine sex hormones in ovine, also approved by University of Florida IACUC (#202111333).

### 2.2. Validation Study Outline

Our study was designed to assess the analytical performance of the IMMULITE^®^ 2000 XPi progesterone assay (Siemens, Cary, NC, USA) in ovine serum. Five of the nine immunoassay validation studies recommended by the guidelines of the Quality Assurance and Laboratory Standard committee of the ASVCP [15] were used. Our approach consisted of two complementary phases that inform both precision and bias aspects of method validation. Specifically, the spike-recovery phase provided data on reportable range, within run, between run, recovery, while the inter-lab comparison study provided data on systematic bias of identical machines in two different labs.

### 2.3. Progesterone Assay via Colorimetric, Chemiluminescent Based Immunoassay (CLEIA)

The IMMULITE^®^ 2000 P4 assay kit (Siemens, Cary, NC, USA) is manufactured for use in Siemens closed system, the IMMULITE^®^ 2000 XPi automated analyzer. The intended use per manufacture’s product insert documentation is in vitro diagnostics to measure serum P4 levels. The basis of P4 quantification is a solid-phase, competitive chemiluminescent enzyme immunoassay. System calibration verifiers and lot specific adjustors are based in human serum and were performed in accordance with manufacturer recommendations, six month and two-week intervals, respectively. Commercially available quality control samples (Lyphocheck, Bio-Rad, Hercules, CA, USA) were used as assay positive controls each day in accordance with manufacture’s recommendations. Each of three quality control (QC) levels were run in triplicate and considered passing if CV% for each level was below 10%. The quality control manufacturer reported mean values for P4 concentration were 0.64, 7.5 and 20.9 ng/mL for QC levels low, medium and high concentrations, respectively.

Default system calibration verifiers set the calibrated range for serum P4 concentration at 0.2 to 40 ng/mL. All samples were analyzed using the same Siemens P4 kit, lot number D585, including the inter-lab study in which the external testing facility reported the same lot number.

### 2.4. Serum Spiking

Spiked serum samples were used for reportable range (linearity), repeatability (within-run precision), intermediate precision (between-run precision), recovery, and as positive controls in the inter-laboratory (reproducibility) studies. Serum was separated by centrifugation of whole blood samples collected from four rams and pooled to serve as the P4 free serum matrix. Serum was stored at −20 °C for a maximum of 48 h before use.

Commercially available certified reference material for P4 at a concentration of 1 mg/mL in acetonitrile (Cerilliant^®^ Sigma, St. Louis, MO, USA) was used as the source of P4. A working stock of P4 was made at 1000 ng/mL and was diluted to final spiked concentrations of 0.5, 1, 2, 5, 10, 15, 20, 30, and 40 ng/mL with the pooled serum as the diluent matrix. Pooled serum was the matrix control. Subsequent sections refer to the concentrations as L1, L2, L3, L4, L5, L6, L7, L8, L9, L10 for 0, 0.5, 1, 2, 5, 10, 15, 20, 30, and 40 ng/mL, respectively; 0 ng/mL corresponds to the serum matrix control. The range of spike concentrations used correlates to the manufacture’s reported calibration range, 0.2 to 40 ng/mL. All measurements using spiked samples were used within 5 days of preparation, and samples were stored at −20 °C.

### 2.5. Reportable Range (Linearity) Study

Five within run replicates of each concentration were performed on day 1. Averages for each concentration were calculated. The association between measured means and the spiked concentrations was assessed using a scatter plot and simple linear regression was performed. Paired *t*-test was used for a rough estimation of the difference between measured concentration and spike concentration data.

### 2.6. Precision

Precision is a measure of how accurately a method performs under independent conditions. These conditions can explain repeatability (variations within run), and reproducibility (variations between runs) [19]. Due to difficulty in mathematical quantification of precision, the inverse, imprecision, is reported using the standard deviation and coefficient of variation percent.

### 2.7. Within-Run (Repeatability)

To test the repeatability of the system across the reportable range, five within-run replicates from each spiking level were used. Standard deviation and coefficient of variation percentages (CV%) was calculated at each level. CV% were plotted against the spiked concentrations and trend lines were generated.

### 2.8. Between-Run (Reproducibility)

The averages of five replicates for each spiking concentration over five consecutive days were used to test the reproducibility of the system. The average standard deviation and average CV% were calculated at each level. Average CV% were plotted against the spiked concentrations and trend lines were generated.

### 2.9. Recovery Study

The Spiking-recovery bias percentage (SRB) was calculated using the formula: SRB (%) = %R − 100
where percent recovery (%R) was calculated for each concentration (L2 to L9; denoted “L_x_” in the formula) using the formula:%R = (L_x_ − L1)/L_spike concentration_) × 100

Although P4 concentration in the matrix control was reported as below detectable limits, 0.1 was used as L_1_ for this calculation to account for potential endogenous P4 and minimize over or underestimates in recovery calculations. The average of five replicates from day 1 was used as the corresponding L_x_ value. The L10 was reported >40 ng/mL, outside of machine reportable range and therefore %R was not calculated. The SRB (%) for each spiked concentration was plotted on a function graph with SRB (y-axis) against spiked concentrations (x-axis).

### 2.10. Inter-Lab Comparison

To assess the average bias (AB), and the range-based bias (RB) across various concentrations between two laboratories, an interlaboratory comparison study was performed. Thirty study samples were collected for an independent study examining circulating P4 concentration in ewes and spike samples were the same samples used in the spiking phase of this study. The P4 concentration ranged from 0.22 to 8.13 ng/mL for study samples. Seven samples were selected in the range 0.22 to 0.7 ng/mL; specifically, 0.28, 0.40, 0.47, 0.51, 0.60, 0.68 and 0.7 ng/mL. Thirteen samples were selected in the range 0.7 to 2.5 ng/mL; specifically, 0.85, 0.88, 0.92, 0.95, 1.0, 1.2, 1.3, 1.3, 1.6, 1.7, 2.0, 2.2, and 2.4 ng/mL. Ten samples were selected in the range 2.5 to 8.1 ng/mL; specifically, 2.6, 2.7, 2.9, 3.2, 3.5, 3.7, 4.1, 5.1, 5.3, and 8.1 ng/mL. These samples (*n* = 39) were analyzed on the IMMULITE^®^ 2000 XPi in our lab using standard protocol settings as set up by the manufacturer including kit lot adjustments and commercial quality controls run as recommended. The samples were then sent to the reference lab, University of Tennessee College of Veterinary Medicine endocrinology diagnostic services. The reference lab was blind to the results from our system as well as sample metadata.

Study samples were stored at −20 °C from collection to initial in house analysis, re-frozen then an aliquot was sent for external analysis. Study samples were collected spring and summer 2021, in house P4 concentration quantification was conducted in fall 2021 and samples were sent for external quantification spring 2022. Spiked samples were stored at −20 °C for two months between in-house and external quantification.

Passing-Bablok regression analysis was used to identify the line of best fit. In Passing-Bablock regression, the intercept value reflects constant bias and the slope value reflects proportional bias. Constant and proportional biases were considered significant when the 95% confidence intervals (CI) did not include zero and one, respectively.

Bland–Altman plots were constructed to characterize the agreement between the two laboratories. The mean bias and percentage bias and associated 95% CI were calculated [20]. The mean constant bias describes the differences between the two laboratory analyzers being consistently above or below 0. Percent bias describes the increase or decrease in the difference between the two laboratories’ readings in proportion to a relative increase in the mean value. Linear regression was used to assess the proportional error in Bland–Altman plots [20,21]. Upper and lower limits of agreement (LoA) were calculated using the following equation:Mean bias = ±1.96 × SD

The plot was visually examined to determine if the differences were symmetrically distributed around 0 (homoscedastic) and that 95% of the differences were between the upper and lower LoA [22]. The IMMULITE^®^ 2000 XPi system was appropriate for measuring P4 concentration in the sample matrix when the 95% CI for LoA < Tea [23]. Statistical analyses were performed using MedCalc Statistical Software version 20.110 (MedCalc Software bvba, Ostend, Belgium, 2018), R Studio version 4.1.3, and SAS® OnDemand for Academics (PROC HPSPLIT; SAS Inst, Inc., Cary, NC, USA). Statistical significance was set at *p* < 0.05.

### 2.11. IMMULITE^®^ 2000 XPi P4 Assay Total Error Computation

Total observed error (TEo) considers coefficient variation and bias. It is calculated using the formula described by Harr et al. [16]: TEo (%) = 2CV% + absolute value of bias %

Absolute value of bias (ASVCP) defines bias as the difference between a measured concentration and a known concentration; it can be expressed as a bias (%) using the following formula described by Harr et al. [16]:Bias (%) = ([mean_target_ − mean_measured_/mean_target_] × 100)

The type of bias that is calculated determines the kind of TEo that can be determined. We calculated TEo based on spiking-recovery bias (SR), range-based bias (RB) and average-based bias (AB).

Specifically:TEo_SR_ (%) = 2CV% + absolute SRB%(1)
where CV% is equal to that calculated for within run precision at each spiking concentration and SRB% is equal to that calculated for recovery study.
TEo_RB_ (%) = 2CV% + absolute difference bias %(2)
where CV% is equal to that calculated for within run precision at each spiking concentration and the difference bias % from the Bland–Altman plot of each group. Ranges were set to encompass the range of biological samples used for the comparison study. A total of 40 samples were organized in increasing concentration then divided into 5 groups for analysis (Table 1).
TEo_AB_ = 2CV% (within-run) + absolute difference bias %(3)
where CV% is equal to that calculated for within run precision at each spiking concentration and total average difference bias % from the Bland–Altman plot of the comparison study.

## 3. Results

### 3.1. Spike and Recovery Phase 

The P4 free matrix used in the spiking samples was consistently reported below the manufacturer calibration range of 0.2 ng/mL. The 40 ng/mL spike sample consistently reported >40 ng/mL both in-house and at the external testing facility, therefore it was excluded from subsequent calculations.

Our data supports a reportable range for P4 concentration of 0.5 to 30 ng/mL with linear association with the spiked serum P4 concentration (R^2^ = 0.99; Figure 1). The regression formula Y = a + bX was used to evaluate the linear regression; “Y” is the observed values as measured by the IMMULITE^®^ 2000 XPi unit, “a” is y-intercept, “b” is the slope and “X” is the spiked P4 concentration. Specifically, the y-intercept was calculated as −0.44 (close to 0; *p* = 0.402), and the slope was calculated as 1.28 (*p* < 0.001) indicating the absence of constant bias and the presence of proportional difference. The paired t-test for linearity indicates a significant deviation from linearity (*p* < 0.001). Precision of this system was inferred by calculating the imprecision of within and between run experiments. Within run data provides a measure of repeatability. As described in materials and methods five replicates were tested for each spike sample. Average within-run CV% over the measurable range was 4.4%. The general trend was for CV to decrease with increasing P4 concentration with the highest calculated CV% at P4 concentration at spiked concentrations 0.5 and 1 ng/mL of 7.2% and 7%, respectively, then slightly increase again at 30 ng/mL at 4% (Figure 2A). However, CV% within-run imprecision at spike concentration 5 ng/mL was higher than lower spike concentration 2 ng/mL, CV% calculated at 6.7% and 2.2%, respectively; this evened out in the between run analysis. Additionally, CV% for spike concentration 30 ng/mL trended upward at 4% for within-run imprecision calculation compared to spike concentrations 10, 15, and 20 ng/mL with CV% 3.2%, 2.2% and 2.7%, respectively. Between-run data provides a measure of reproducibility. As described in materials and methods between run imprecision was calculated across the measurable range over five consecutive days using CV%. Average between-run CV% over the measurable range was 4.3%, again following the trend of decreasing CV% with increasing spike P4 concentration, until 20 ng/mL then a slight rise for spike concentration 30 ng/mL (Figure 2B). Specifically, the highest imprecision was noted at spike concentration 0.5 ng/mL and the lowest at 20 ng/mL at 8% and 2%, respectively. At spike concentration 15 ng/mL elevated imprecision was observed at 4% compared to 3% and 2% for spike concentrations 10 and 20 ng/mL, respectively (Table 1).

To inform accuracy of this system SRB% was calculated as described in materials and methods. The highest absolute SRB% was 38% for spike concentration 0.5 ng/mL, while the lowest was 1.5% for spike concentration 5 ng/mL (Table 1). No overall trend was observed for absolute SRB% over the reportable range (Figure 3). Absolute SRB% were 38, 6, 22, 1.5, 11, 22, 33, and 26% for spike concentrations 0.5, 1, 2, 5, 10, 15, 20, and 30 ng/mL, respectively (Table 1).

### 3.2. Inter-Lab Comparison Phase

An inter-lab comparison study was conducted to assess reproducibility between systems as described in materials and methods. Passing–Bablok regression analysis was used to measure systematic and proportional bias between the two machines. Analysis was conducted for the readable range (0.2 to 30 ng/mL) including both study and spike samples as described in materials and methods. Graphical representation of this analysis is presented in Figure 4. The intercept is −0.11 with 95% confidence interval 0.05 to 0.16. The slope is 1.14 with a 95% confidence interval of 1.12 to 1.17. Both parameters suggest there is constant difference between the two systems; the 95% CI of the intercept does not contain 0 nor does the 95% CI of the slope contain 1 [24]. Spearman rank correlation coefficient is 0.995 with *p* < 0.0001 and 95% confidence interval of 0.991 to 0.997. Cusum test for linearity had a *p* = 0.50, indicating samples were evenly represented across the testing concentration.

Bland–Altman analysis was used to measure the agreement between the two units [25]. The data from this analysis suggests that agreement between the two units is not consistent, as average concentration increases difference between the two units is exaggerated (Figure 5). The P4 quantified at the external testing facility was numerically less than in house system for all samples tested, furthermore, as the P4 increases this discrepancy in quantification increases. Although most data points are between the ±1.96 standard deviation, they are not randomly scattered around the mean; suggesting P4 in the sample influences the agreement between the two units.

Spike recovery total error was calculated as described in materials and methods for each spike concentration. The average TEo(SR) was 28.73% with the highest being at 0.5 ng/mL (52.36%) and the lowest at 5 ng/mL (14.81%) (Figure 6, Table 1). Average TEo(RB) and (AB) were 29.3% and 33.3%, respectively. The highest TEo(RB) was observed at 0.5 ng/mL (59.5%) and the lowest was at 15 ng/mL (19.1%). TEo(AB) was more even across the reportable range. The highest TEo(AB) was at 0.5 and 1 ng/mL (38.9% and 38.6%. respectively); the lowest TEo(AB) was at 2 and 10 ng/mL (28.9%).

## 4. Discussion

The data reported here indicates the manufacturer’s default protocol on the closed IMMULITE^®^ 2000 XPi system provides accurate and reliable quantification of progesterone in ovine serum. This is the first report to thoroughly investigate analytical performance parameters of this closed CLEIA system in sheep serum. 

The manufacturer’s reportable P4 concentration measurement range of the IMMULITE^®^ 2000 XPI P4 immunoassay (IPI) is 0.2–40 ng/mL. This means that, according to manufacturer internal testing, the system measures P4 concentration from 0.2 to 40 ng/mL with acceptable precision and accuracy. Herein, the reportable range of the IPI was evaluated by linearity using spike samples. These data confirm that the reportable range provided by the manufacturer is achievable in ovine serum. Visual inspection of the data indicates excellent linearity as indicated by the R^2^ value as well as a consistent over-estimate of P4 concentration in ovine serum, especially at concentrations > 5 ng/mL as indicated by deviation from the true line (Figure 2). 

Precision information was inferred from imprecision measurements (CV%). Current bioanalytical method validation guidelines from multiple U.S. federal services, including the Department of Health and Human Services and Center for Veterinary Medicine indicate a CV of ±20% as acceptable for both within- and between-run ligand binding assays [26]. In this study, IPI showed excellent repeatability and reproducibility both with CV < 10% for each concentration. Average within- and between-run CV% was <5 over the whole reportable range. As expected, the lower concentrations, 0.5 and 1 ng/mL, have higher CV%; 7.2 and 7.0, respectively for within-run and 8 and 6, respectively for between-run; Standard deviation is below 0.1. Despite the numerically higher CV% for the clinically relevant 1 ng/mL concentration, it remains well within the ±20% acceptable variance [26]. Therefore, this automated system is precise enough to be used in research and clinical settings for measuring serum P4 concentration in sheep. The precision of IMMULITE^®^ 2000 XPi reported in here is similar to previous findings for cow serum P4 concentration using the IMMULITE^®^ 1000 [27].

All unit specific analytical performance parameters tested here using spiked samples fall within acceptable method variance [28] supporting use of this machine to measure ovine circulating P4 concentration over the reportable range. Of note, the progesterone used in these experiments was certified reference P4 material, any currently undescribed differences between this reference and ovine progesterone could add a source of bias in the spiking recovery phase of experiments. However, any variation is unlikely given the high level of conservation [29]. Additionally, analysis of P4 concentration from the patient samples is consistent with expected results based on age, sex and pregnancy status, using the clinical parameter of luteal phase P4 concentration >1 ng/mL [5]. Also notable is that the experimental design did not include evaluation of serum matrix cross-reactivity components. Dehydroepiandrosterone (DHEA) is known to cause falsely elevated progesterone results in immunoassays [30], however, DHEA is reported to be low in sheep [31]. While the presence of P4 mimics could contribute to falsely elevated readings, the use of P4 free sheep serum matrix as described in materials and methods suggests that any potential cross-reactivity does not impact the interpretation of this validation data because it was the same across all spiking concentrations, thus impacting each reading equally. 

In addition to precision, Passing–Bablok and Bland–Altman analysis were used to mathematically determine if the quantified concentration in the inter-laboratory phase were interchangeable. The data from our analysis suggest that the absolute quantification result obtained for ovine serum P4 concentration is not the same for two independent systems. Passing–Bablok regression analysis provides a mathematical model to determine if two methods are interchangeable by assessing both systematic and proportional bias [24]. Two parameters of this analysis that inform the practical interpretation of interchangeability of two methods is the intercept and the slope; if the 95% CI of the intercept and the slope include 0 and 1, respectively there is no systematic or proportional bias and thus the methods can be used interchangeably [24]. Our data suggests that quantification data obtained from our IMMULITE^®^ 2000 XPi system is not interchangeable with that from another IMMULITE^®^ 2000 XPi system as neither parameter was satisfied (95% CI for intercept was 0.05042 to 0.1567 and for slope was 1.1159 to 1.1712). Additionally, Bland–Altman analysis is used to mathematically explain agreement between two systems was used. The Bland–Altman analysis across measurable range shows that the difference between units does not cluster around the mean, in fact visual examination shows an increasing positive bias with increasing concentration indicating extensive manipulations would be necessary to compare quantification data across units. However, the data points until for concentrations < 15 ng/mL fall between the + 1.96 standard deviations indicating observed differences would not have clinical implications [32]. Taken together Passing–Bablok and Bland–Altman analysis indicate that although absolute quantification of P4 concentration is not reproducible between the two systems tested data bias around the sheep clinically relevant 1 ng/mL falls within acceptable deviation implying numerical differences in measured concentration between machines will not change the clinical interpretation. 

An additional strength of this study is the calculation of three types of biases (SR, RB, and AB) and three types of TEo (TEo_SR_, TEo_RB_, or TEo_AB_). Each type of bias has different properties and provides a certain type of information, for example, SR bias reflects the interior bias of the IPI over the spiked P4 concentration range, RB and AB shows the bias between two laboratories. Range-based bias is considered the most relevant type of bias in the clinical setting because it shows the bias between two laboratories at a given P4 concentration, but the AB averages all of RB biases which minimizes the true bias [13]. The average TEo of IPI reported in this study was ~30%. This means that samples with exactly P4 concentration of 1 ng/mL, have a 95% probability of being measured within 1 ng/mL ± TEo which is 0.8 to 1.2 ng/mL. This percentage of the total error is acceptable based on several recommendations [15,26,28]. However, TEo is highest at lower concentrations, and at 1 ng/mL calculated biases considered acceptable only by the most lenient recommendation (<40%) [26].

## 5. Conclusions

In conclusion, the data presented here suggests the use of the IMMULITE^®^ 2000 XPi closed system is a reliable, precise method to measure ovine serum P4 concentration for clinical and research purposes. However, use of absolute concentration maybe slightly variable and not interchangeable between facilities. This study provides important information about the precision and accuracy of IPI that should be considered in the interpretation of results and for future expert consensus discussions to determine the recommendations for allowable total error (TEa).

## Figures and Tables

**Figure 1 animals-12-03534-f001:**
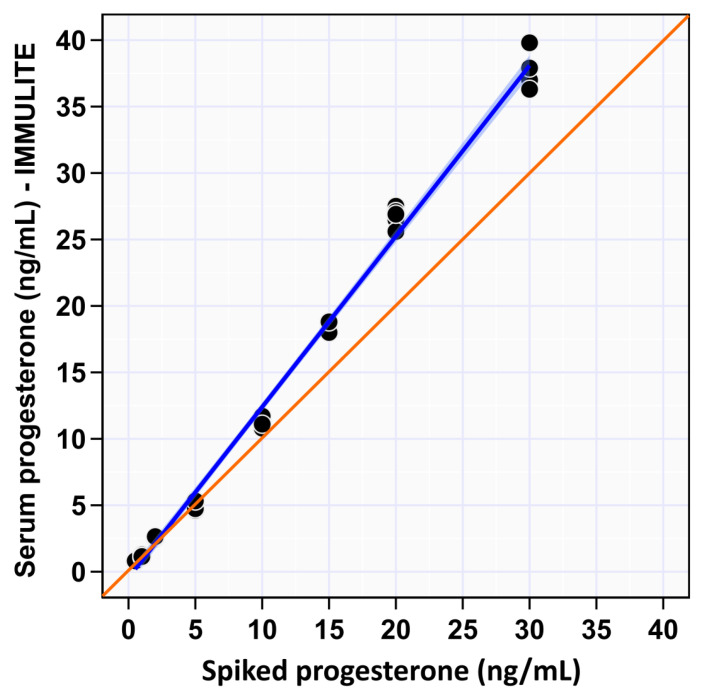
Scatter plot showing linearity of reportable range of spiked serum P4 concentration measured by IMMULITE^®^ 2000 XPi immunoassay (Siemens, Cary, NC, USA). The orange line is identity line, and the blue line is the regression line. Regression line equation calculated in Excel: True P4 = −0.44 + 1.28x.

**Figure 2 animals-12-03534-f002:**
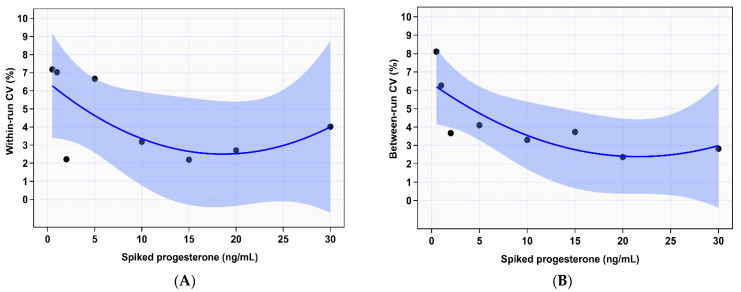
Within-run (**A**) and between-run (**B**) precision of IMMULITE^®^ 2000 XPi P4 immunoassay across the spiked serum progesterone concentrations reported as the coefficient of variation percentage (CV%). The shaded blue area indicates 95% CI of mean difference. Dark blue line indicates the fitted regression line.

**Figure 3 animals-12-03534-f003:**
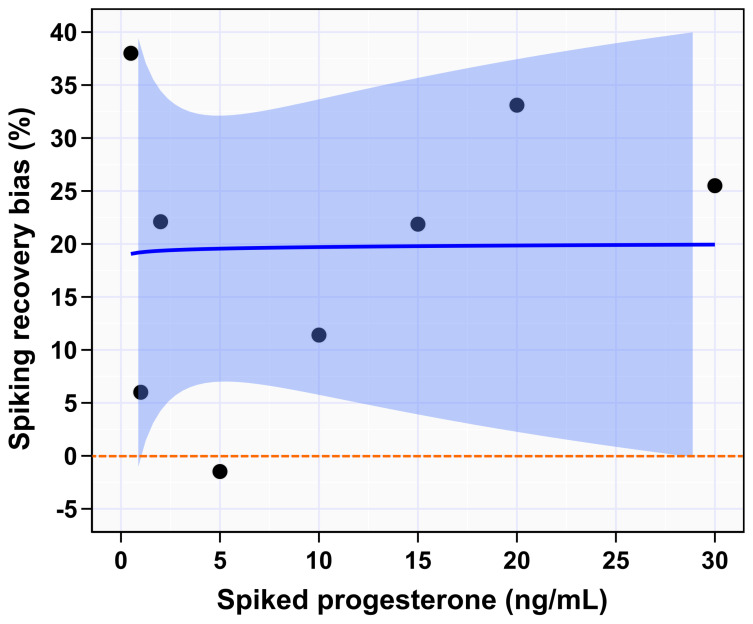
Spiking-recovery bias percentage of IMMULITE^®^ 2000 XPi P4 immunoassay across the spiked serum. The shaded blue area indicates 95% CI of mean difference. Dark blue line indicates the fitted regression line.

**Figure 4 animals-12-03534-f004:**
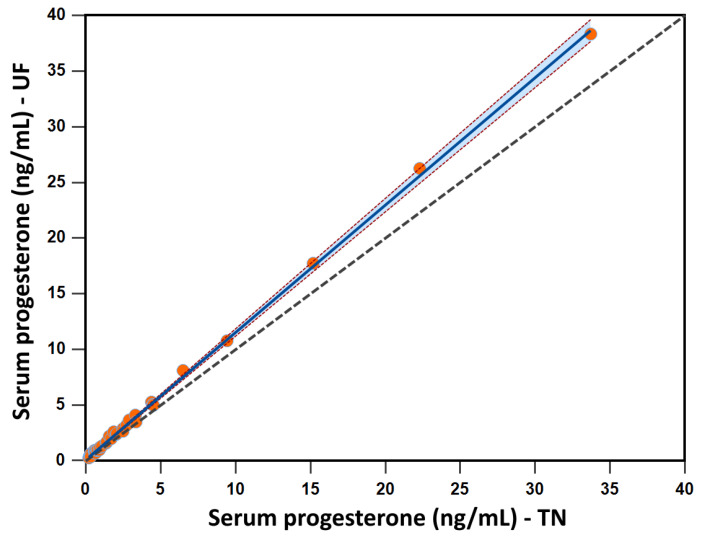
Scatterplot indicating the relationship between the serum progesterone concentration measured by the IMMULITE^®^ 2000 XPi P4 immunoassay at the University of Florida (UF) and the University of Tennessee (TN). The grey dashed diagonal line is the line of identity, and the solid blue line is the line of best fit from Passing–Bablok.

**Figure 5 animals-12-03534-f005:**
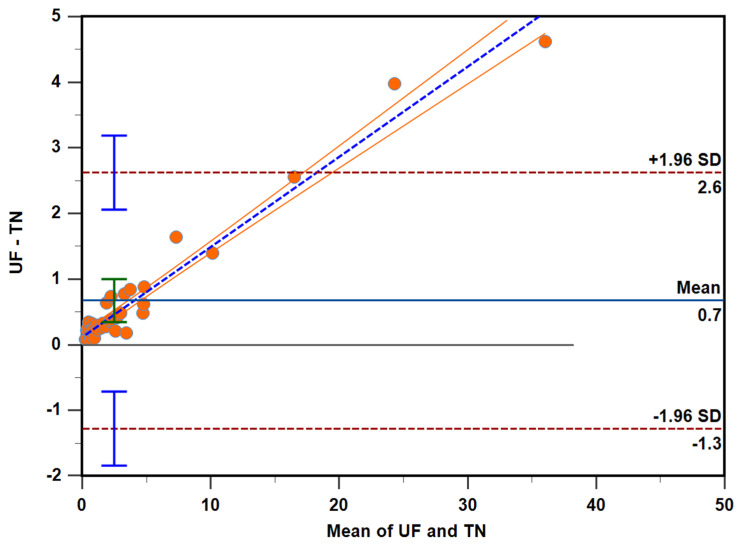
Bland–Altman mean difference plot between the serum progesterone concentration measured by IMMULITE^®^ 2000 XPi P4 immunoassay at the University of Florida (UF) and the University of Tennessee (TN). The horizontal blue solid line represents the mean bias and the horizontal brown long dashed lines reflect the 95% limits of agreement. The horizontal gray solid line represents the line of identity. The blue short dashed line and orange solid lines indicate the fitted regression line and the 95% CI, respectively. The blue and green vertical bars represent the 95% CI.

**Figure 6 animals-12-03534-f006:**
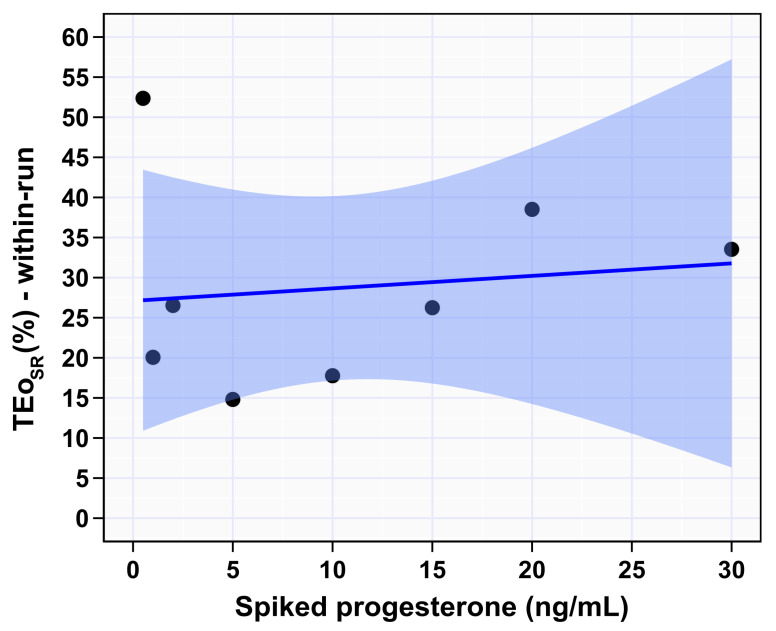
Observed total error percentage (TEo) of IMMULITE^®^ 2000 XPi progesterone immunoassay across the spiked serum. The shaded blue area indicates 95% CI of the mean difference. The dark solid blue line indicates the fitted regression line.

**Table 1 animals-12-03534-t001:** IMMULITE^®^ 2000 XPi P4 assay precision and total observed error results for readable range in spiked sheep serum.

P4 Concentration (ng/mL)		Precision (%CV)		Total Observed Error (%)		
Spiked	Measured	Within	Between	TEo (SR)	TEo (RB)	TEo (AB)
P4	P4	run	run			
0.5	0.8	7.2	8.1	52.4	59.5	38.9
1	1.2	7.0	6.3	20.1	36.7	38.6
2	2.5	2.2	3.7	26.5	26.0	28.9
5	5.0	6.7	4.1	14.8	28.8	37.8
10	11.2	3.2	3.3	17.8	21.3	30.9
15	18.4	2.2	3.7	26.2	19.3	28.9
20	26.7	2.7	2.4	38.5	20.3	29.9
30	37.8	4.0	2.8	33.5	22.9	32.5
average		4.4	4.3	28.7	29.3	33.3

## Data Availability

Please contact the corresponding author for data requests.

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
