# Peer review of "Analytical Validation of the IMMULITE® 2000 XPi Progesterone Assay for Quantitative Analysis in Ovine Serum"

_animals, 2022, doi:10.3390/ani12243534_

Round 1

Reviewer 1 Report

Manuscript ID animals-2060063  

I had the opportunity to review the manuscript entitled “Analytical validation of the Immulite® 2000 XPi progesterone assay for quantitative analysis in ovine serum”.   

In this study, the authors characterise the analytical performance of the IMMULITE® 2000 XPi Progesterone (P4) immunoassay across the reportable range in sheep serum.   

The analytical performance evaluated included the determination of linearity, precision through within-run, and between-run coefficient of variation (CV) calculations, accuracy through bias calculations for spiking-recovery bias and interlaboratory (range and average based) for two laboratories across the reportable range (0.2 – 40 ng/mL)..  

The results obtained were satisfactory, based on these data; the authors conclude that the IMMULITE® 2000 XPi P4 automated immunoassay provides a precise, accurate, reliable and safe method for the measurement of [P4] in sheep serum.  

The paper, in line with the topic of the journal, is of interest to the scientific community, well written and easy to read, with many graphs and figures; it needs a few minor additions.  

Below my minor considerations line-by-line  

L. 126: “Each of three QC levels …”; I suggest inserting the meaning of the acronym “QC”  

L. 145-147: check the sentence “Subsequent sections refer to the concentrations as L1, L2, L3, L4, L5, L6, L7, L8, L9, L10 for matrix control and 0.5, 1, 2, 5, 10, 15, 20, 30, and 40 ng/mL respectively.”  

L 180: check the formula “%R = (Lx-L1l) / Lspike concentration) * 100”. 

L 194-196: I suggest reporting the physiological values of P4 concentration in blood given in the current references.  

L. 268: check and rewrite “(R2=0.99. Figure 1)”. 

L. 313, 355: in figure 3 and figure 6, insert “blue area represented 95% CI of mean difference”.  

L. 362: “the IPI is 0.2-40 ng/mL.” …”; I suggest inserting the meaning of the acronym “IPI”.

References: check references n. 24 and 26. 

24. U.S. Department of Health and Human Services; Food and Drug Administration; Center for Drug Evaluation and Research (CDER); Center for Veterinary Medicine (CVM) Bioanalytical Method Validation Guidance for Industry. (Add month and year)  

26. Cox, K.L.; Devanarayan, V.; Kriauciunas, A.; Manetta, J.; Montrose, C.; Sittampalam, S. Immunoassay Methods; 2004; (2004 in bold)  

Author Response

Manuscript ID animals-2060063

Reviewer 1

I had the opportunity to review the manuscript entitled “Analytical validation of the Immulite® 2000 XPi progesterone assay for quantitative analysis in ovine serum”. In this study, the authors characterize the analytical performance of the IMMULITE® 2000 XPi Progesterone (P4) immunoassay across the reportable range in sheep serum. The analytical performance evaluated included the determination of linearity, precision through within-run, and between-run coefficient of variation (CV) calculations, accuracy through bias calculations for spiking-recovery bias and interlaboratory (range and average based) for two laboratories across the reportable range (0.2 – 40 ng/mL). The results obtained were satisfactory, based on these data; the authors conclude that the IMMULITE® 2000 XPi P4 automated immunoassay provides a precise, accurate, reliable and safe method for the measurement of [P4] in sheep serum. The paper, in line with the topic of the journal, is of interest to the scientific community, well written and easy to read, with many graphs and figures; it needs a few minor additions. Below my minor considerations line-by-line

  1. 126: “Each of three QC levels …”; I suggest inserting the meaning of the acronym “QC”

Author response: Thank you for pointing out this oversight. We have updated the sentence to read “Each of three quality control (QC) levels were run in triplicate and considered…”

  1. 145-147: check the sentence “Subsequent sections refer to the concentrations as L1, L2, L3, L4, L5, L6, L7, L8, L9, L10 for matrix control and 0.5, 1, 2, 5, 10, 15, 20, 30, and 40 ng/mL respectively.”

Author response:  Thank you for pointing out this confusing sentence, we have re-written it to be more clear that L1 is the serum matrix control not 0.5 ng/mL. It has been updated to read “Subsequent sections refer to the concentrations as L1, L2, L3, L4, L5, L6, L7, L8, L9, L10 for 0, 0.5, 1, 2, 5, 10, 15, 20, 30, and 40 ng/mL respectively; 0 ng/mL corresponds to the serum matrix control.”

L 180: check the formula “%R = (Lx-L1l) / Lspike concentration) * 100”.

Author response: Thank you for pointing out this mistake. The formula has been updated to “ %R=(Lx-L1)/Lspike concentration)*100 “

L 194-196: I suggest reporting the physiological values of P4 concentration in blood given in the current references.

Author response: Thank you for this suggestion. We agree and have included these specific values and the section pertaining to lines 194-196 now reads “The [P4] ranged from 0.22 to 8.13 ng/mL for study samples. Seven samples were selected in the range 0.22 to 0.7 ng/mL; specifically, 0.28, 0.40, 0.47, 0.51, 0.60, 0.68 and 0.7 ng/mL. Thirteen samples were selected in the range 0.7 to 2.5 ng/mL; specifically, 0.85, 0.88, 0.92, 0.95, 1.0, 1.2, 1.3, 1.3, 1.6, 1.7, 2.0, 2.2, and 2.4 ng/mL, Ten samples were selected in the range 2.5 to 8.1 ng/mL; specifically, 2.6, 2.7, 2.9, 3.2, 3.5, 3.7, 4.1, 5.1, 5.3, and 8.1 ng/mL.”

  1. 268: check and rewrite “(R =0.99. Figure 1)”.

Author response: Thank you for bringing this oversight to our attention. We have removed the “.” and the sentence now reads “Our data supports a reportable range for [P4] of 0.5 to 30 ng/mL with linear association with the spiked serum [P4] (R2=0.99; Figure 1).”

  1. 313, 355: in figure 3 and figure 6, insert “blue area represented 95% CI of mean difference”.

Author response: Thank you for pointing out that this clarification is necessary. The sentence has been added to both figure legends.

  1. 362: “the IPI is 0.2-40 ng/mL.” …”; I suggest inserting the meaning of the acronym “IPI”.

Author response:  Thank you for pointing out this oversight. We have updated the sentence to include the meaning of the acronym and the sentence now reads: “The manufacturer’s reportable [P4] measurement range of the IMMULITE® 2000 XPI P4 immunoassay (IPI) is 0.2-40 ng/mL.”

References: check references n. 24 and 26.

  1. U.S. Department of Health and Human Services; Food and Drug Administration; Center for Drug Evaluation and Research (CDER); Center for Veterinary Medicine (CVM) Bioanalytical Method Validation Guidance for Industry. (Add month and year)
  2. Cox, K.L.; Devanarayan, V.; Kriauciunas, A.; Manetta, J.; Montrose, C.; Sittampalam, S. Immunoassay Methods; 2004; (2004 in bold)

Author response: Thank you, the references have been updated and now appear in the manuscript as indicated below:

  1. U.S. Department of Health and Human Services; Food and Drug Administration; Center for Drug Evaluation and Research (CDER); Center for Veterinary Medicine (CVM) Bioanalytical Method Validation Guidance for Industry. May 2018

  1. Cox, K.L.; Devanarayan, V.; Kriauciunas, A.; Manetta, J.; Montrose, C.; Sittampalam, S. Immunoassay Methods; 2004.

Reviewer 2 Report

In this manuscript entitled “Analytical validation of the Immulite 2000 VPi progesterone assay for quantitative analysis in ovine serum”, the authors test progesterone concentrations in sheep serum and evaluate the product used in this study.

Although the experimental procedures and interpretation of the data are of overall good quality, the reviewer is unable to consider whether this kit has an accurate and reliable quantification of progesterone in sheep serum.

Comments:

1. The "Introduction" should further elaborate on the significance of progesterone, with appropriate citations. In addition, it is also necessary to clarify how good the quality of progesterone detection by previous detectors has been and where the problems lie.

2. All experiments lacks appropriate (both positive and negative) controls. Test data from sera collected under various conditions, etc. are needed.

3. To demonstrate that the Immulite 2000XPi immunoassay kit can truly show reliable quantitative results,  it is necessary to examine it from a multifaceted analysis, including analysis by mass spectrometer. 

4. Cross-reactivity of other steroid hormones must be addressed to demonstrate accurate measurement of progesterone.

Author Response

Reviewer 2

In this manuscript entitled “Analytical validation of the Immulite 2000 VPi progesterone assay for quantitative analysis in ovine serum”, the authors test progesterone concentrations in sheep serum and evaluate the product used in this study. Although the experimental procedures and interpretation of the data are of overall good quality, the reviewer is unable to consider whether this kit has an accurate and reliable quantification of progesterone in sheep serum.

Author note: English revisions were conducted.

Comments:

  1. The "Introduction" should further elaborate on the significance of progesterone, with appropriate citations. In addition, it is also necessary to clarify how good the quality of progesterone detection by previous detectors has been and where the problems lie.

Author response: We agree and thank you for pointing out this missing critical piece of information in the introduction. To address this, we have edited the first paragraph to include this information and added the corresponding references and the first paragraph now reads: “Progesterone (P4) is a conserved steroid hormone [1] and concentration ([P4]) plays a role in both establishment and maintenance of pregnancy [2–4]. Circulating P4 can be measured in serum or plasma and levels are both cyclical and age dependent. Specifically, in female sheep, >1ng/mL circulating [P4] indicates luteal phase of cycling[5].  Therefore circulating [P4] can be used to determine puberty and inform reproductive management practices in female small ruminants [6]. As sheep extensive production systems increase to aid in global food security, multiple challenges including reproduction need to be studied and addressed to best optimize sustainability of these systems [7]. Timed artificial insemination and embryo transfer are some examples of methods used in reproduction management, and it has been shown that circulating [P4] at various times can influence the success of these techniques in sheep [8]. As research and reproductive management in small ruminants progresses it is essential that methods used to measure [P4] in blood components are reliable, accurate, safe, timely, and cost effective. The historical lab standard to measure circulating P4 levels is radio-immunoassay, however, its use is limited by time constraints, cost, safety and regulatory concerns [9]. Therefore, the use of colorimetric, chemiluminescent based immunoassay (CLEIA) has been increasing, and fully automated CLEIA systems are now widely used in human clinical settings to measure circulating [P4]. The Immulite® 2000 XPi (Siemens. Cary, NC) is one of these automated systems. It is a closed system designed to measure a variety of analytes, including P4 in a multiplex array from multiple bodily fluid sources. This machine has been designed and validated for use in human samples.  However, although there are reports of using the Immulite® 2000 XPi, and its predecessor the Immulite® 1000 to measure circulating [P4] in veterinary setting[10–12], there are no comprehensive analytical performance reports addressing the accuracy or reliability of this data in ruminants. This is especially important as reproductive and clinical demands are increasing, specifically in small ruminants as a result of progressive producer interest in breeding practices.”

  1. All experiments lacks appropriate (both positive and negative) controls. Test data from sera collected under various conditions, etc. are needed.

Author response: Thank you for this critique. To clarify controls, we have re-worded L130 in materials and methods to indicate the commercially available quality controls run daily were the positive controls. This sentence now reads “Commercially available quality control samples (Lyphocheck. Bio-Rad. Hercules, CA) were used as assay positive controls each day in accordance with manufacture’s recommendations.”. Negative controls were the serum free samples that consistently reported values below the calibrated range as indicated in the results section L272-273. The goal of our validation study was to evaluate the analytical parameters of the machine so people using it are aware of the systematic bias (as is present in any assay/method and indicated in the introduction section, L99-104) the use of spiked samples of known concentration is the appropriate approach and patient samples from various conditions falls out of the scope of this experimental objective. This is supported by ASVCP QALS Guideline, Version 3.0 (2019) as referenced in materials and methods L117.

  1. To demonstrate that the Immulite 2000XPi immunoassay kit can truly show reliable quantitative results, it is necessary to examine it from a multifaceted analysis, including analysis by mass spectrometer.

Author response: Thank you for this thought-provoking critique. The objective of our study was to validate the machine within the parameters of the method itself to determine if the application of this method yields acceptable results to measure progesterone in sheep serum based on known input concentrations. Our omission of method comparison study is in line with previously published validations studies:

  1. Cate FL, et al. Analytical and clinical validation of the Immulite 1000 hCG assay for quantitative analysis in urine. Clin Chim Acta. 2013 Jun 5;421:104-8. doi: 10.1016/j.cca.2013.02.026.
  2. Korchia J, Freeman KP. Validation study of canine urine cortisol measurement with the Immulite 2000 Xpi cortisol immunoassay. J Vet Diagn Invest. 2021 Nov;33(6):1052-1068. doi: 10.1177/10406387211031194.
  3. Korchia J, Freeman KP. Validation study of canine serum cortisol measurement with the Immulite 2000 Xpi cortisol immunoassay. J Vet Diagn Invest. 2021 Sep;33(5):844-863. doi: 10.1177/10406387211029247.

  1. Cross-reactivity of other steroid hormones must be addressed to demonstrate accurate measurement of progesterone

Author response: Thank you for this insightful critique. We have now addressed this in the discussion by including the following: “Also notable is that the experimental design did not include evaluation of serum matrix cross-reactivity components. Dehydroepiandrosterone (DHEA) is known to cause falsely elevated progesterone results in immunoassays [30], however, DHEA is reported to be low in sheep [31]. While the presence of P4 mimics could contribute to falsely elevated readings, the use of P4 free sheep serum matrix as described in materials and methods suggests that any potential cross-reactivity does not impact the interpretation of this validation data because it was the same across all spiking concentrations, thus impacting each reading equally.”   L402-409

Reviewer 3 Report

The present work describes a series of analyses to validate Immulite® 2000 XPi progesterone assay, initially designed for human samples in sheep samples. The methodology is simple and the results and discussion are succinct. In a few words, the authors describe and demonstrate the validation of an interesting tool that now can be applied in sheep production and medicine. Although, the results might not be an easy read for outside-of-the-field readers I recommend the paper for publication.

Author Response

Reviewer 3

The present work describes a series of analyses to validate Immulite® 2000 XPi progesterone assay, initially designed for human samples in sheep samples. The methodology is simple and the results and discussion are succinct. In a few words, the authors describe and demonstrate the validation of an interesting tool that now can be applied in sheep production and medicine. Although, the results might not be an easy read for outside-of-the-field readers I recommend the paper for publication.

No comments for author to address.

Round 2

Reviewer 2 Report

All the reviewers' concerns were addressed.